# Analysis of Phenolic Compounds of *Reynoutria sachalinensis* and *Reynoutria japonica* Growing in the Russian Far East

**DOI:** 10.3390/plants13233330

**Published:** 2024-11-27

**Authors:** Andrey R. Suprun, Konstantin V. Kiselev, Olga A. Aleynova, Artem Yu. Manyakhin, Alexey A. Ananev

**Affiliations:** Federal Scientific Center of the East Asia Terrestrial Biodiversity, Far Eastern Branch of the Russian Academy of Sciences, 690022 Vladivostok, Russia

**Keywords:** Japanese knotweed, Reynoutria, HPLC-MS/MS, stilbenes, phenolic acids, flavan-3-ols, flavones, naphthalene, anthraquinones, phenylpropanoid disaccharide esters

## Abstract

The Russian Far East is a region of unique biodiversity, with numerous plant species, including *Reynoutria japonica* and *Reynoutria sachalinensis*. These plants are considered a serious threat to biodiversity and are classified as threatened species. However, *Reynoutria* plants synthesize and accumulate a variety of metabolites that are valued for their positive effects on human health. The main objective of this study is to quantitatively and qualitatively evaluate the content of secondary metabolites in different parts of *R. japonica* and *R. sachalinensis* plants. In this study, the results of phylogenetic analysis of the *ITS2*, *matK*, and *rps16* genes showed that samples collected in the Sakhalin region were closest to *R. sachalinensis*, while samples collected in Primorsky krai were closer to *R. japonica*. The high-performance chromatography and mass spectrometry (HPLC-MS/MS) method was used to identify the compounds. As a result of the identification of metabolites in the leaves, stem, and roots of *R. japonica* and *R. sachalinensis*, we showed the presence of a total of 31 compounds, including stilbenes, phenolic acids, flavan-3-ols, flavones and flavonols, naphthalene derivatives, anthraquinones and derivatives, and phenylpropanoid disaccharide esters. The root of *R. japonica* was shown to be a rich source of stilbenes (up to 229.17 mg/g DW), which was 8.5 times higher than that of *R. sachalinensis* root (up to 27.04 mg/g DW). The root also contained high amounts of emodin derivatives and vanicoside B. Quercetin and its derivatives were the major metabolites in the leaves and stems of both *Reynoutria* species. In *R. japonica* leaves, quercetin-3-*O*-pentoside was the major compound, reaching a total of 7 mg/g DW, accounting for 34% of all compounds analyzed. In contrast, in *R. sachalinensis* leaves, quercitrin was the major compound (up to 13.96 mg/g DW), accounting for 62% of all compounds and 12.7 times higher than in *R. japonica* leaves. In turn, *R. japonica* leaves also contained high amounts of phenolic acids (up to 10 mg/g DW). Thus, the obtained results showed significant differences in the qualitative and quantitative composition of metabolites between *R. japonica* and *R. sachalinensis* plants. Additionally, in this work, a cell culture of *R. japonica* was obtained and tested for its ability to synthesize and accumulate stilbenes.

## 1. Introduction

Japanese knotweed sensu lato is a group of perennial rhizomatous herbs in the family Polygonaceae, genus *Reynoutria* Houtt. [1,2]. These plants are endemic to East Asia but have been introduced to the West, where they have become invasive and resilient weeds [3]. This group includes two main species: Japanese knotweed (*Reynoutria japonica* (Houtt.), synonym *Fallopia japonica*) and giant knotweed (*Reynoutria sachalinensis* (F. Schmidt), synonym *Fallopia sachalinensis*), as well as hybrids between the two species [3]. *R. japonica* can also be further classified into a number of related species or intraspecific taxa, most of which are endemic to East Asia. However, two of them occur outside this range: the tall plain form var. japonica (*R. japonica* s.s.) and the dwarf mountain form var. compacta (Hook.f.) [4]. These plants are considered a serious threat to biodiversity and are classified as dangerous invasive species, including the list of the 100 most dangerous invasive species by the International Union for Conservation of Nature (IUCN) [5]. Due to its ability to thrive in various soil types and environments, it poses a significant threat and has the potential for widespread distribution [3]. The knotweed can spread abundantly both asexually (through rhizomes and adventitious roots) and sexually (through seeds) [4]. The spread of knotweed is also significantly aided by its tendency to inhabit coastal areas and disperse seeds both by water and by air [6]. The rhizomes play a significant role in the spread of knotweed, allowing it to reproduce through fragmentation and clonal growth. This enables the plant to rapidly colonize new areas and overshadow other species, contributing to its invasive nature [7].

Despite its parasitic nature, Japanese knotweed is widely used in clinical practice in East Asian countries such as China, Japan, and Korea. In China, it is officially included in the Chinese pharmacopoeia and is often prescribed by practicing doctors for the treatment of hepatitis, jaundice, amenorrhea, burns, bruises, hyperlipidemia, and cough [8]. In addition to its medical applications, Japanese knotweed is often used in everyday food in both China and Japan. The root part of *Fallopia multiflora* (Thunb.) is used in Korea to prepare beverages [9]. *Polygonum cuspidatum* is also used for therapeutic purposes, although less frequently due to its narrower range of therapeutic effects [10,11]. Various parts of *R. japonica* are used in traditional Chinese medicine due to their therapeutic effects on various inflammatory diseases and their immune-stimulating properties [12]. It is known that various parts of the Japanese knotweed plant are rich in various secondary metabolites, including flavonoids (rutin, apigenin, quercetin, isorhamnetin, and kaempferol), stilbenes (resveratrol, polydatin, resveratroloside, piceatannol), anthraquinones (emodin, citreorosein, fission, fallacinol, chrysophanol, phylloquinones B and C, and anthraglycosides A and B), coumarins, essential oils, and others (lapatoside, 8-hydroxycalamenene, oleanolic acid, chlorogenic acid, protocatechuic acid, gallic acid, tachioside, etc.) [13,14,15,16,17]. Among the numerous plants that synthesize stilbenes, Japanese knotweed is noted for having one of the highest contents of resveratrol (up to 12.1 mg/g DW) [18], polydatin (up to 40.3 mg/g DW) [19], resveratroloside (up to 19.6 mg/g DW) [14], piceatannol glucoside (up to 1.22 mg/g DW) [20], and astringin (up to 1.92 mg/g DW) [21]. These compounds have been thoroughly studied and are believed to play a crucial role in the medicinal properties of Japanese knotweed [10]. In addition to medicinal and gastronomic use, the effectiveness of *Reynoutria* sp. extracts in improving plant growth and protecting against pathogenic microorganisms has been proven. Extracts from *R. japonica*, *R. bohemica*, and *R. sachalinensis* generally showed an improvement in wheat germination and growth of *Triticum aestivum* wheat plants [22]. Moreover, it has been shown that *R. sachalinensis* extract is able to reduce the incidence of powdery mildew in tomato and cucumber plants [23,24]. In the study by Anžlovar et al. 2020, *F. japonica* extracts had a broad spectrum of low to medium antifungal activity against *Epicoccum nigrum* and *Fusarium poae* [25]. Thus, the efficacy of *Reynoutria* sp. extracts indicates its potential as a disease control agent and growth and development promoter for agricultural plant species.

The Russian Far East is a region with unique biodiversity, rich in numerous plant species, including Japanese knotweed (*R. japonica* and *R. sachalinensis*). Previous studies show that the content of secondary metabolites, such as stilbenes, varies significantly. However, data on the metabolites of Japanese knotweed growing in the southeastern part of Russia remain limited, highlighting the need for new research to gain a more comprehensive understanding. The primary aim of this study is to quantitatively and qualitatively assess the content of secondary metabolites in different parts of *R. japonica* and *R. sachalinensis* (leaves, stems, and roots). An additional objective is to develop and optimize methods for obtaining cell cultures of Japanese knotweed for in vitro synthesis of stilbenes. This is important for creating sustainable sources of secondary metabolites that can be used in the pharmaceutical and cosmetic industries.

## 2. Results

### 2.1. Phylogenetic Analysis

A phylogenetic tree of nine members of the genus *Reynoutria* and *Fallopia* and collected samples (Appendix A) was constructed to clarify the species affiliation and taxonomy of the studied samples. The phylogenetic tree was constructed using maximum likelihood analysis based on the sequences of one nuclear gene *ITS* and two chloroplast genes *matK* and *rps16-trnK*. These genes were selected based on literature data [2]. The *matK* and *rps16* gene sequence analysis revealed that samples 1–4 were closely related to *R. sachalinensis*, while samples 5–14 corresponded to *R. japonica* (Figure 1 and Appendix A). Further analysis of the *ITS* sequences revealed that samples 5–14 were also completely identical to *R. japonica*. However, the *ITS* sequences of samples 1–4 differed significantly from the previously described ITS sequences for *R. sachalinensis* (Appendix A). This phenomenon can be explained by intraspecific variation in the *ITS* gene, indicating possible genetic differences within the species. The sequences of *matK* and *rps16* genes, samples 1–4 were completely identical to each other, as well as to previously described sequences of *R. sachalinensis*. Similar results were obtained for samples 5–14, which were identical to *R. japonica* (Appendix A). It is worth noting that the available *R. sachalinensis ITS* sequences in GenBank were collected from plants growing at a significant distance from the samples we studied, such as Korea or Europe. This may indicate geographic variation within the species. There is also a possibility that in Sakhalin Oblast we encountered a subspecies or variation of *R. sachalinensis*, which requires further detailed study. Joint analysis of the three genes showed that samples 1–4 (collected in Sakhalin Oblast, Russia) are closest to *R. sachalinensis*, while samples 5–14 (collected in Primorsky Krai, Russia) are closest to *R. japonica* (Figure 1). Finally, based on the morphological determination and the genetic analysis, we identified the analyzed plants as *R. japonica* and *R. sachalinensis*.

### 2.2. Analysis of Phenolic Compounds by HPLC-MS/MS

As a result of the identification of metabolites in the leaves, stems, and roots of *R. japonica* and *R. sachalinensis*, we showed the presence of a total of 31 compounds, including stilbenes (piceatannol 3′-O-glucoside, resveratroloside, polydatin, resveratrol), phenolic acids (chlorogenic acid, coumaroyl quinic acid, 5-O-caffeoylquinic acid), flavan-3-ols (procyanidin dimer, epicatechin, catechin, epicatechin-3-gallate), flavones and flavonols (luteolin 7-O-glucoside, quercetin-3-O-glucoside, quercetin-3-O-pentoside, quercitrin, kaempferol-3-O-glucoside, quercetin), naphthalene derivative (torachrysone glucoside, torachrysone), anthraquinones and derivatives (emodin glucoside, emodin-8-O-(6′-O-malonyl)-d-glucoside, hydropiperoside, physcion-8-O-β-d-glucoside, questin, emodin) and phenylpropanoid disaccharide esters (vanicoside B and vanicoside C) (Figure 2; Appendix A). It is worth noting that some substances, such as polydatin and coumaroyl quinic acid, were found in multiple isoforms. Compounds were identified by comparison with their standards, mass spectra (MS/MS), retention time (Rt), and their UV spectra reported in the literature [17,26,27,28,29].

We analyzed 10 plants collected in Primorsky Krai (*R. japonica*) and 8 plants collected in Sakhalin Oblast (*R. sachalinensis*). Samples from Primorsky Krai had a common spectrum of metabolites, the amount of which varied insignificantly. A similar effect was obtained for *R. sachalinensis* samples from Sakhalin Oblast. In different parts of *R. japonica* and *R. sachalinensis* plants, both common and specific metabolites characteristic of each individual species were found. A similar effect was obtained for *R. sachalinensis* samples. Based on this, we combined the metabolite content data for each species and tissue type and presented them in Table 1 and Table 2. The results we obtained may indicate the adaptation of plants to their habitat, and can also be used as biomarkers for species identification. In turn, the presence of common metabolites may indicate genetic proximity between species.

The stilbenes profiles of *R. japonica* and *R. sachalinensis* were similar, but the contents of these compounds varied considerably (Table 1). The highest total stilbenes content was shown for the root of *R. japonica* (up to 229.17 mg/g DW), which was 8.5 times higher than in the root of *R. sachalinensis* (up to 27.04 mg/g DW). The main stilbenes in the root of both species were polydatin, resveratroloside, and piceatannol 3′-O-glucoside (Table 1). The leaves and stems of both species contained predominantly trans-polydatin, the amount of which was 4.8 and 2 times higher in the leaves and stem of *R. sachalinensis*, respectively, compared to *R. japonica*. Moreover, the leaves of *R. sachalinensis* contained cis-polydatin, although in small amounts (Table 1). The large number of stilbenes in the root can be explained by the fact that the root part of the plant grows and accumulates stilbenes over many years, while the aboveground part exists only during one growing season.

We have shown that *R. japonica* leaves contain a significant amount of phenolic acids (up to 10 mg/g DW), while phenolic acids were not detected in *R. sachalinensis* samples. It is known that coumaroyl quinic acid occurs in three isoforms (isomer of 758, 759, 760) [23], the total amount of which reached 3.1 mg/g DW, i.e., one third of the total amount of phenolic acids in leaves. The content of chlorogenic acid (3-O-caffeoylquinic acid) in *R. japonica* leaves reached 4.8 mg/g DW, and 5-O-caffeoylquinic acid up to 2 mg/g DW (Table 2).

Additionally, differences in the metabolite profile were noted for substances related to flavan-3-ols, namely, catechin and its derivatives and procyanidin B, which was detected only in the root of *R. sachalinensis* (up to 0.48 mg/g DW). In all samples of plants *R. japonica*, only epicatechin was detected, the content of which varied from trace amounts in leaves to 0.39 mg/g DW in the stems. In contrast, epicatechin was not detected in the tissues of *R. sachalinensis*, but the presence of epicatechin-3-gallate was shown, the maximum amount of which was in the leaves (up to 1.1 mg/g DW), where catechin was also present (up to 3.6 mg/g DW) (Table 2).

The next group identified in *R. japonica* and *R. sachalinensis* plants were flavones and flavonols, which included seven compounds that were derivatives of quercetin, luteolin, and kaempferol (Figure 2). Luteolin 7-O-glucoside was found in the leaves and stems of both species, mainly in trace amounts (less than 0.1 mg/g DW). All samples also showed a fairly low amount of kaempferol-3-O-glucoside, no more than 0.7 mg/g DW in the roots of *R. sachalinensis* (Table 2). Quercetin and its derivatives were more abundant in the leaves and stems of both *Reynoutria* species. In the leaves of *R. japonica*, the main compound was quercetin-3-O-pentoside, the content of which reached a total of 7 mg/g DW, which represented 34% of all analyzed compounds. In contrast, in the leaves of *R. sachalinensis*, the main compound was quercitrin (up to 14 mg/g DW), which accounted for 62% of all compounds and was 12.7 times higher than in the leaves of *R. japonica*. In both studied species, the content of flavones and flavonols in the stems and roots was low, not exceeding 2 mg/g DW in total (Table 2).

We have shown that one of the key groups of compounds found in the plants *R. japonica* and *R. sachalinensis* are anthraquinones and derivatives, which are represented by six compounds (emodin glucoside, emodin-8-O-(6’-O-malonyl)-d-glucoside, hydropiperoside, physcion-8-O-β-d-glucoside, questin, emodin), some of which are contained in fairly large quantities, especially in the roots (Figure 2). Based on our data, the largest proportion in plant tissues is accounted for by emodin glycosylated forms, namely, emodin glucoside and emodin-8-O-(6’-O-malonyl)-d-glucoside, the content of which was 9.71 mg/g DW and 5.22 mg/g DW, respectively, in the roots of *R. japonica*, which is 12 and 6.4 times higher than in the root part of *R. sachalinensis* (Table 2). Emodin glucoside was also contained in the leaves of *R. japonica* and *R. sachalinensis*, 2.47 mg/g DW and 1.18 mg/g DW, respectively. Questin was detected in trace amounts in the stems and small amounts in the roots of *R. sachalinensis* (up to 1.36 mg/g DW) (Table 2). Physcion-8-O-β-d-glucoside was detected only in *R. sachalinensis* leaves.

Vanicoside C and vanicoside B are phenylpropanoid disaccharide esters and are known to be found in Japanese knotweed plants [17], which is confirmed by this study. We detected these compounds in the stems and roots of both studied species of *Reynoutria*. One of the most represented substances in the roots of both species is vanicoside B. For example, its share in the roots of *R. japonica* is 7%, and in the roots of *R. sachalinensis*, it is 34% of the total amount of analyzed metabolites (Table 2). The content of vanicoside B in the stems of *R. japonica* reached 3.94 mg/g DW, and in the stems of *R. sachalinensis* 14.34 mg/g DW, which is 3.6 times more (Table 2). The amount of vanicoside C was maximum in the roots of *R. japonica* and *R. sachalinensis* and amounted to 6.24 mg/g DW and 6.49 mg/g DW, respectively (Table 2).

The last group of compounds detected in Japanese knotweed plants was the naphthalene derivative, which includes substances such as torachrysone and its glycoside torachrysone glucoside, which has been described only for the roots of *R. sachalinensis* (0.06 mg/g DW). Torachrysone was contained in the stems up to 0.378 mg/g DW and 0.52 mg/g DW, and roots up to 1.84 mg/g DW and 2.09 mg/g DW of *R. japonica* and *R. sachalinensis*, respectively (Table 2).

### 2.3. Stilbene Accumulation in Cell Cultures of R. japonica

In this study, we have shown that *R. japonica* plants are a rich source of stilbenes. Callus cell culture is a promising system for stilbenes production. Therefore, we were tasked with obtaining a cell culture of *R. japonica* and testing it for its ability to synthesize and accumulate stilbenes. The highest level of stilbenes was detected in the roots of *R. japonica* (up to 229.17 mg/g DW), but we were unable to obtain a cell culture from this part of the plant due to the high degree of fungal and bacterial contamination. However, we successfully obtained a cell line from the leaves (Figure 3b). The resulting callus cells were dense, slow-growing homogeneous tissues synthesizing a small amount of stilbenes on MS nutrient medium containing 0.5 mg/L 6-benzylaminopurine and 2 mg/L α-naphthaleneacetic acid (Table 3). Interestingly, the cell culture accumulated mainly stilbenes, namely trans-polydatin and trans-resveratrol, 0.33 mg/g DW and 0.02 mg/g DW, respectively (Figure 3a,c). It was suggested that optimization of the amount of phytohormones is required to initiate stilbenes synthesis in *R. japonica* cell cultures. Since auxins are known to be one of the hormones that stimulate cell division, we used different concentrations of α-naphthaleneacetic acid. For control, we also used MS nutrient medium without phytohormones. It was shown that the minimum amount of stilbenes was produced on the nutrient medium without phytohormones (0.17 mg/g DW) (Table 3). The use of medium (2 mg/L) and high (4 mg/L) concentrations of 6-benzylaminopurine did not lead to an increased stilbene content, and was 0.35 mg/g DW and 0.39 mg/g DW. The use of 6-benzylaminopurine in a low concentration (0.5 mg/L) led to a faster weight gain, but the amount of stilbenes still remained low (0.42 mg/g DW). Moreover, cultivation of *R. japonica* cell culture for six months did not result in degeneration into loose, actively growing homogeneous tissues.

## 3. Discussion

Determining the species affiliation of Reynoutria plants is sometimes a difficult task. Moreover, the taxonomic classification of knotweed has changed many times since its initial classification, and in the literature, authors often use the genera Polygonum, Reynoutria, and Fallopia as species epithets [3]. At present, several varieties of *R. japonica* have been described, such as *R. japonica* var. *japonica*, *R. japonica* var. *hachidyoensis*, and *R. japonica* var. *uzenensis* [1]. All are endemic to Japan and only *R. japonica* var. *japonica* is known to have been introduced outside its native range. *R. sachalinensis*, on the other hand, is native to Japan, Sakhalin, and Ullung-do (an island between Korea and Japan) [1]. However, the relationship of some morphotypes, which mostly originate from the area where the distribution ranges of *R. sachalinensis* and *R. japonica* overlap geographically, is unclear. The fact is that knotweed has a higher level of genetic diversity in its native range than in its invasive range [30]. Additionally, in the natural range, there are many more subspecies and varieties of knotweed compared to the adventive range [3,30]. In this work, we showed that the nucleotide sequences of the *rps16* and *matK* genes of *R. japonica* and *R. sachalinen* were identical for the analyzed plants of *R. japonica* and *R. sachalinen*, respectively. However, the *ITS* gene sequences of the plants growing on Sakhalin were only 93% similar to the previously described *ITS* sequences of *R. sachalinen*. This phenomenon can be explained by intraspecific variability of the *ITS* gene, which indicates possible genetic differences within the species. Moreover, there is also a possibility that in the Sakhalin region we encountered a subspecies or variation of *R. sachalinensis*, which requires further detailed study.

Japanese knotweed is a rich source of available polyphenols such as stilbenes, flavonoids, coumarins, essential oils, and others [13,14,15,31]. The large variety and number of diverse substances explains its positive effects on human health, such as positive effects on endocrine and cardiovascular systems, antitumor, antiviral, antioxidant, and anti-pulmonary fibrotic effects [10,32]. One of the most well-known and widely studied groups of substances that knotweed contains is the stilbenes. Various plant species from more than 30 families are known to synthesize stilbenes, but most of them are not able to accumulate these metabolites in large quantities. To the best of our knowledge, high levels of stilbenes have been reported in the bark of *Picea jezoensis* (250 mg/g DW) [33] and *Pinus koraiensis* (54.8 mg/g DW) [34], in the bark of *Morus albus* 54 mg/g DW [35], and in the root of *Vitis vinifera* 10.8–10.9 mg/g DW [36]. The roots of knotweed plants are known to contain high amounts of resveratrol, piceid, and other stilbenes [20,29]. Vastano et al. 2020, showed that *Polygonum cuspidatum* roots predominantly accumulated piceatannol glucosid, resveratroloside, piceid, and resveratrol, the sum of which co-accumulated 12.6 mg/g DW [20]. Similar results were obtained by Nawrot-Hadzik et al. 2018, which showed that the roots of *R. japonica* contained 14.83 mg/g piceid and 1.29 mg/g resveratrol. Moreover, the authors did not detect stilbenes in the rootstock of *R. sachalinensis* [29]. However, there are studies in which resveratrol content reached up to 12.1 mg/g DW [18], polydatin up to 40.3 mg/g DW [19], and resveratroloside up to 19.6 mg/g DW [14], which is comparable to some of the highest values of stilbenes in all plants. In this study, it was shown that both *R. japonica* and *R. sachalinensis* are capable of synthesizing and accumulating stilbenes. We showed that the stilbenes content in the roots of *R. japonica* reached 229 mg/g DW, and in the roots of *R. sachalinensis* up to 27 mg/g DW. The major stilbenes in the roots of both species were polydatin, resveratroloside, and piceatannol 3′-O-glucoside, which is in general agreement with other studies [18,19,20]. The leaves and stems of both species contained predominantly trans-polydatin, but in lower amounts compared to the root part of the plant. The results indicate that *R. japonica* is a rich source of stilbenes along with other plant species.

In addition to stilbenes, knotweed plants are a source of phenolic acids, but these substances in amounts up to 10 mg/g DW were detected only in the leaves of *R. japonica*. However, phenolic acids were not detected in *R. sachalinensis* samples. This absence may indicate differences in metabolism and biochemical profile between the two species, which may be related to their different adaptation to environmental and ecosystem conditions. It was previously shown that shoots of *R. japonica* produce more phenolic acids than roots, with the highest amount of phenolic acids obtained in shoots from ex vitro plants (11.35 mg/g DW) [28]. Lachowicz and Oszmiański 2019 showed that the content of phenolic acids in *F. japonica* was 1.1 times higher than in *F. sachalinensis*, reaching 0.66 g/100 g DW [27]. Thus, in our study, the co-number of phenolic acids was comparable to that reported by other studies. It is known that phenolic compounds are synthesized by plants in response to stress factors [28]. Accordingly, depending on the growth conditions of knotweed plants, the amount of phenolic acids may vary.

The next widespread group of substances in plants, including Japanese knotweed, is flavan-3-ols, which includes catechin and its derivatives. Previously, Lachowicz et al. 2019 showed that the concentration of flavan-3-ols in leaves and rhizomes of *R. japonica* plants can reach 88% of all phenolic compounds. They also reported 11.64 g/100 g flavan-3-ols in samples in *F. japonica*, which was 1.2 times higher than that in *F. sachalinensis* samples [27]. Plants of *R. japonica* grown under in vitro conditions accumulated catechin and its derivatives from 5 to 20 mg/g DW depending on the cultivation conditions, but not the plant part difference [28]. The content of catechin and epicatechin in young shoots of different knotweed species has also been previously studied in detail. It was shown that *R. japonica* and *R. sachalinensis* accumulate mainly epicatechin, 568 and 674 mg/kg DW, respectively [37]. In this study, we showed that epicatechin was present in all samples of *R. japonica* plants, with contents as high as 0.39 mg/g DW in the stems. Epicatechin was not detected in *R. sachalinensis* tissues, but epicatechin-3-gallate was shown to be present, with a maximum amount in leaves (up to 1.1 mg/g DW), where catechin was also present (up to 3.6 mg/g DW).

Many plants, including *R. japonica* and *R. sachalinensis*, are known to synthesize various flavones and flavonols, such as quercetin, kaempferol, luteolin, and their derivatives [18,27,32]. These substances have a wide range of biological activities [12]. We showed that in *R. japonica* leaves, quercetin-3-O-pentoside was the main compound up to 7 mg/g DW, accounting for 34% of all co-compounds analyzed. In contrast, in *R. sachalinensis* leaves, the main compound was quercitrin up to 14 mg/g DW, which accounted for 62% of all compounds and was 12.7 times higher than in *R. japonica* leaves. Lachowicz and Oszmiański 2019 showed that flavones and flavonols were mainly in leaves and accounted for 12% of total phenolics, but in stems they accounted for 5%, and in roots 0.2% of total phenolic compounds [27]. The chemical composition of leaves, stems, and roots of *P. multiflorum* also differed [38].

Naphthalene, torachrysone, and torachrysone glucoside are classified as naphthalene glycosides that were first discovered in *Rumex obtusifolius* [39]. These compounds are common in the Polygonaceae family and have attracted considerable interest due to their diverse biological activities including α-amylase inhibition, antiviral, and anti-inflammatory effects [40]. Naphthalene glucoside has been described in the roots of *R. japonica* growing in Korea at 1.22 mg/g DW. However, in the same study, naphthalene derivative was not detected in *R. sachalinensis* plants, but was detected in *R. forbesii* at 3.72 mg/g DW [14]. In this study, we showed that *R. japonica* and *R. sachalinensis* plants contained torachrysone mainly in the root part. It was shown that its amount in roots reached 1.84 mg/g DW and 2.09 mg/g DW of *R. japonica* and *R. sachalinensis*, respectively, which is consistent with the data in the scientific literature [14,27].

Anthraquinones are an important class of compounds widely distributed in plants, including plants of the Polygonaceae family. Various anthraquinone compounds such as emodin, physcion, questin, and their derivatives have been reported to possess various pharmacological activities such as antioxidant, antimicrobial, anti-inflammatory, and anticancer [40,41]. According to the information available in the literature, emodin, physcion, and glycosides of 9,10-anthraquinone derivatives were found in both rhizomes and aboveground parts of *R. japonica* and *R. sachalinensis* [17,18,29,31,42]. Nawrot-Hadzik et al. 2018 showed that emodin content reached 4.93 mg/g DW in *R. japonica* plants, compared to *R. sachalinensis* 0.13 mg/g DW and *R. x bohemica* 1.93 mg/g DW. The same study showed that physcion content reached 2.96 mg/g DW in *R. japonica* plants, 0.19 mg/g DW in *R. sachalinensis,* and 1.86 mg/g DW in *R. x bohemica* [29]. A higher content of anthraquinones was described for *P. cuspidatum* roots, where the amounts of emodin and physcion reached 6.09 mg/g and 2.26 mg/g, respectively. Our study showed that anthraquinones and their derivatives are one of the key groups of compounds found in plants of *R. japonica* and *R. sachalinensis*, which is represented by six compounds (emodinglucoside, emodin-8-O-(6′-O-malonyl)-d-glucoside, hydropiperoside, physcion-8-O-β-d-glucoside, questin, emodin), some of which are found in rather large amounts, especially in roots. Emodin glucoside and emodin-8-O-(6′-O-malonyl)-d-glucoside have the largest share in the roots of *R. japonica* and *R. sachalinensis*, the content of which in *R. japonica* was higher than in the roots, 12 and 6.4 times higher than in *R. sachalinensis*. Thus, our results confirm the conclusions of previous studies that the rhizomes of *R. sachalinensis* contain a lower amount of anthraquinones, which, together with a large amount of hydroxycinnamic glycosides, makes it easily distinguishable from the other two knotweed species [29]. However, interestingly, we also detected physion-8-O-β-d-glucoside only in the leaves of *R. sachalinensis*, which may also be a species-specific metabolic trait.

The last group identified in plants of *Reynoutria* species is phenylpropanoid disaccharide esters (vanicoside B and vanicoside C). Vanicoside B is known to be able to exhibit pro-antitumor activity, in particular, to inhibit the two-step skin tumor carcinogenesis in mice induced by 12-O-tetradecanoylphorbol-13-acetate (TPA), and has shown antitumor activity against a panel of cancer cell lines in MDA-MB-231 triple negative breast cancer (TNBC) cells by affecting cyclin-dependent kinase 8 (CDK8) [43]. Vanicoside C and hydropiperoside were first detected in HZ by reverse-phase diode array high-performance liquid chromatography-tomography and time-of-flight mass spectrometry [29]. Vanicoside B is the major form of phenylpropanoid disaccharide esters in *R. japonica* and *R. sachalinensis* plants, and its amount was 0.64 mg/g and 2.25 mg/g, respectively [29]. In this study, we also show that the proportion of vanicoside B in *R. japonica* roots is 7%, and in *R. sachalinensis* roots, it is 34% of the total metabolites analyzed. This makes it one of the most represented substances in the roots of both species.

Establishment of cell or tissue culture of *R. japonica* is a possible direction for further research aimed at increasing the synthesis of secondary metabolites using biotechnological tools. Various authors have described plant cell culture as an efficient and highly productive system for producing a variety of secondary metabolites [28,44]. In this paper, we used *R. japonica* leaves as an explant for callus induction in medium, and analyzed the callus biomass accumulation rate and the ability to produce stilbenes. The roots of *R. japonica* plants are one of the main sources of stilbenes, which have proven to be beneficial to human health [13]. However, some authors indicate that *P. cuspidatum* is difficult to cultivate in callus, due to the difficulty in controlling the combination of hormones and in optimizing the cultivation conditions [45]. In this work, we were also unable to obtain non-flattened, fast-growing tissues synthesizing large amounts of stilbenes on MS solid nutrient medium. Moreover, we were not able to optimize the amount of auxins and cytokines to improve growth performance. Although it has been previously reported that benzylaminopurine (BAP) and α-naphthylacetic acid (NAA) can be used to achieve efficient biomass accumulation and stilbenes production [45,46,47], it is possible that a more detailed analysis of the effect of various phytohormones on the growth process and production of secondary metabolites in cell culture *R. japonica* is needed. Makowski et al. 2024 showed that stirred cultures and the use of bioreactors increased biomass accumulation compared to cultures on solid medium. Moreover, tissue cultures of *R. japonica* had increased synthesis of phenolic compounds compared to plants from ex vitro conditions [28]. Based on our results and literature data, it can be concluded that the use of liquid, stirred cultures is the best strategy to increase the biomass production of Japanese knotweed.

## 4. Materials and Methods

### 4.1. Plant Material and Cell Cultures

Samples of 10 *R. japonica* plants (leaves, stems, roots) were collected in Primorsky Krai, Russia in September 2023. Samples of 8 *R. sachalinensis* plants (leaves, stems, roots) were collected in Sakhalin Oblast, Russia in September 2023 (Appendix A). The roots were extracted together with the soil and placed in a sterile bag; then, the roots were cleaned from the soil in the laboratory. The leaves and stem were collected from the same plant as the root. Each sample was delivered to the laboratory in sterile bags within one day at 10 °C and used for DNA extraction and sample preparation for HPLC. The average temperature and precipitation at the collection site in Primorsky Krai were 19 °C and 50 mm in September 2023 (https://world-weather.ru/pogoda/russia/vladivostok/september-2023 (accessed on 1 September 2024)). In the area of collection points in Sakhalin Oblast, the average temperature and precipitation in September 2023 were 21 °C and 150 mm (https://world-weather.ru/pogoda/russia/yuzhno_sakhalinsk/september-2023 (accessed on 1 September 2024)). A total of 14 biological replicates of *R. japonica* and *R. sachalinensis* were collected and analyzed. In addition, at least 2 technical replicates were used for each biological replicate.

A callus culture was obtained from leaf samples of wild *R. japonica* plants collected in autumn (September) of 2023. After collecting the leaves, they were thoroughly washed in running water. Then they were rinsed with distilled water. The leaves were fragmented with a sterile scalpel into fragments of 0.5 × 0.5 cm. Then, the obtained explants were sterilized with 80% ethanol solution for 90 s and transferred to a 10% aqueous solution of commercial bleach for 15 min. After thorough washing with distilled water, the explants were transferred to Murashige and Skoog (MS) WB/A agar medium [48] supplemented with 0.5 mg/L 6-benzylaminopurine (BAP), 2 mg/L α-naphthylacetic acid (NAA), and 5.5 g/L agar. Every month we transplanted the callus culture into fresh nutrient medium. For the experiment, several variations of the nutrient medium were used, namely: without phytohormones; 0.5 mg/L BAP and 2 mg/L NAA;0.5 mg/L BAP and 0.5 mg/L NAA; 0.5 mg/L BAP and 2 mg/L NAA; 0.5 mg/L BAP and 4 mg/L NAA.

### 4.2. DNA Extraction, Sequencing, and Phylogenetic Analysis

Total DNA was isolated from 20 mg of dried leaves of *R. sachalinensis* (samples 1–4) and *R. japonica* (sample 5–14) according to the method described by Echt et al. 1992 [49], with some modifications [50]. DNA concentration was measured using a spectrophotometer NanoPhotometer P 330 (Implen, Schatzbogen, Germany). For PCR amplification, regions of the sequence of one nuclear gene ITS and two chloroplast genes matK and rps16-trnK were selected. Primers were aimed at conserved sequences of the species Reynoutria, between which the variable region was located (Appendix A). The obtained amplicons were isolated from the gel using the Cleanup Mini kit (Eurogen, Moscow, Russia), cloned to pJET1.2/blunt (Fermentas, Vilnius, Lithuania), and sequenced using the Big Dye Terminator Cycle Sequencing Kit v3.1 reagent kit according to the manufacturer’s method on an ABI 310 Genetic Analyzer sequencer (Applied Biosystems, Waltham, MA, USA) at the Federal Scientific Center of Biodiversity, Far Eastern Branch of the Russian Academy of Sciences.

The resulting nucleotide sequences were reviewed and edited using the Clustal W software [51]. Next, for confirmation to taxa and genes, the obtained sequences were blasted in the NCBI database [52]. Sequences are presented in Appendix A. The distance matrix was constructed using the Geneious [53] program based on 72 gene sequences (*ITS2*, *matK*, and *rps16*) belonging to the genus *Reynoutria* and *Fallopia*, 30 of which were obtained from the NCBI genebank database (Appendix A). The phylogenetic tree was constructed using the method maximum likelihood with the MEGA11 program [54], and node support was estimated by resampling the derived trees by bootstrapping—1000 repetitions [55].

### 4.3. Analysis of Secondary Metabolites by HPLC-MS/MS

For HPLC-MS/MS analysis, leaves, stems, and roots of *R. japonica* and *R. sachalinensis* were dried at 60 °C for 24 h and crushed using laboratory mill IKA A 10 basic (IKA Werke GmbH & Co. KG, Staufen im Breisgau, Germany). A mass of 100 mg of crushed tissue was extracted in methanol at 60 °C for 2 h, as described previously [33,56]. After extraction, the extract was purified with Discovery^®^ DSC-18 SPE Tube bed wt. 50 mg, volume 1 mL (Supelco, Bellefonte, PA, USA), and then used for the HPLC-MS/MS analysis.

The identification of secondary metabolites was performed using a 1260 Infinity analytical HPLC system (Agilent Technologies, Santa Clara, CA, USA) coupled to a Bruker HCT ultra PTM Discovery System (Bruker Daltonik GmbH, Bremen, Germany) equipped with an electrospray ionization (ESI) source, as described earlier [33]. The data were obtained in positive and negative ionization modes. HPLC with diode array detection (HPLC–DAD) for the quantification of all compounds was performed using an HPLC LC-20AD XR analytical system (Shimadzu, Kyoto, Japan), as described earlier [57]. Briefly, the chromatographic separation was performed on a Shim-pack GIST C18 column (150 mm, 2.1 nm i.d., 3 μm particle size; Shimadzu, Japan). For the separation of the analyzed compounds, 0.1% formic acid and acetonitrile were used as mobile phases A and B, respectively, with the following elution profile: 0–35 min—0% B; 35–40 min—40% B; 40–50 min—50% B; 50–65 min—100% B. A volume of 3 μL of the sample extract was injected at a column temperature of 40 °C and a flow rate of 0.2 mL/min.

All identified substances were determined using analytical standards, mass spectra, retention time, and UV spectra, and literature data and public databases such as MassBank, HMDB, PubChem, and NIST Mass Spectral Library (Appendix A). The amount of analyzed metabolites was determined by external standard methods using four-point regression calibration curves constructed using analytical standards. Analytical standards chlorogenic acid, 3-O-p-coumaroylquinic acid, 5-O-caffeoylquinic acid, procyanidin b2, epicatechin, catechin, piceatannol, luteolin, polydatin, quercetin, kaempferol, resveratrol, emodin, physcion, and questin were obtained from Sigma-Aldrich (St. Louis, MO, USA). Additionally, torachrysone was obtained from LGS (London, UK), and vanicoside B and vanicoside C were acquired from MCE (Monmouth Junction, NJ, USA). For quantitative determination of glycosylated forms of substances, standards of their aglycones were used. All solvents used in the analysis were of HPLC grade.

### 4.4. Date Analysis

One-way analysis of variance (ANOVA) with Tukey’s paired comparison test at a significance level of *p* = 0.05, performed using Microsoft Office Excel 365, was used to analyze the experimental results. The level of 0.05 was set as the threshold of minimal statistical significance for all analyses performed.

## 5. Conclusions

In conclusion, it is shown that there is a significant difference in the composition of metabolites between *R. japonica* and *R. sachalinensis* plants. The root of *R. japonica* is a richer source of stilbenes, while the leaves of *R. sachalinensis* contain a higher concentration of quercetin derivatives. These and other differences in the metabolic profile may indicate the adaptation of each species to its habitat. Moreover, they emphasize the possibility of using the detected metabolites as biomarkers for species identification. In turn, the presence of common metabolites may indicate a genetic proximity between *R. japonica* and *R. sachalinensis*. However, despite fast growth and danger as a weed, these Reynoutria plants are used in many areas of human activity, such as medicine, cosmetology, agriculture, and even cooking, which means that research aimed at studying these plants requires continuation.

## Figures and Tables

**Figure 1 plants-13-03330-f001:**
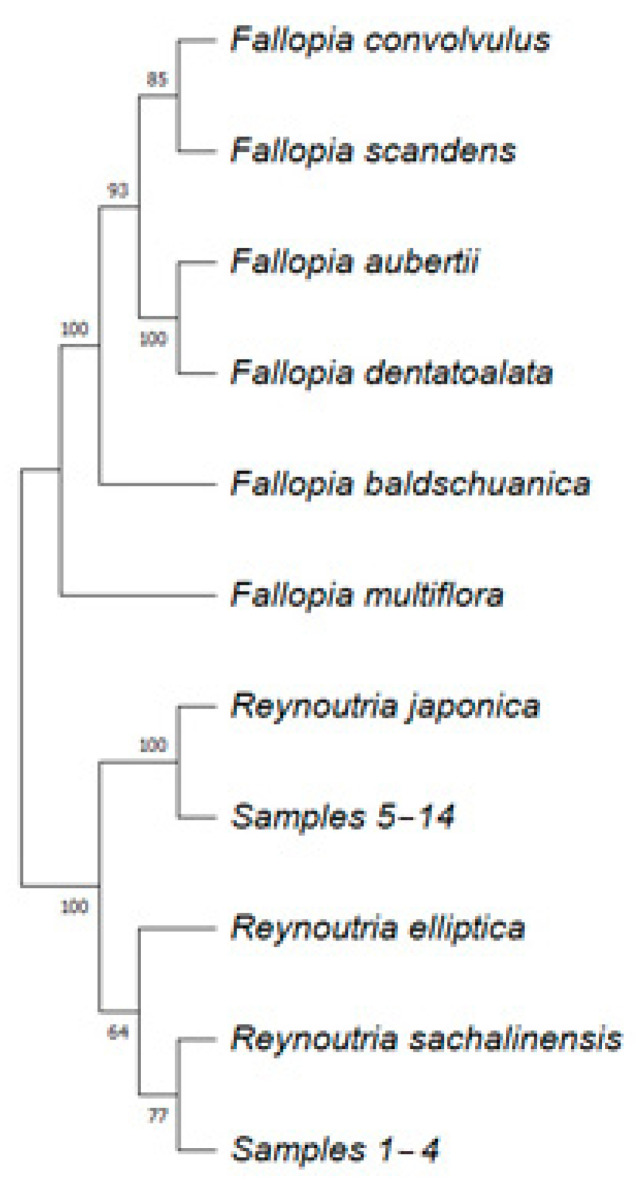
Phylogenetic tree generated by based on maximum likelihood analysis of genes (*ITS2*, *matK*, and *rps16*). Bootstrap support values (≥50%). Samples 1–4 (collected in Sakhalin Oblast, Russia) and samples 5–14 (collected in Primorsky Krai, Russia).

**Figure 2 plants-13-03330-f002:**
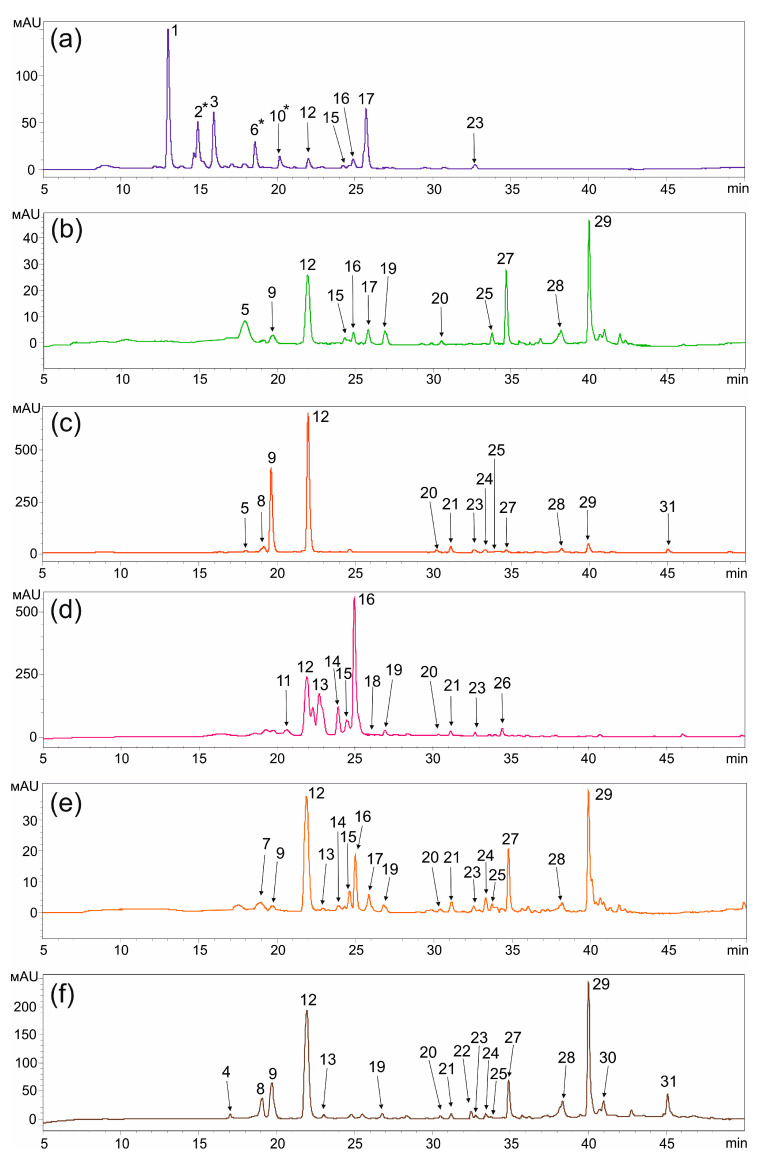
Comparison of HPLC-UV chromatographic profiles (310 nm) for the methanolic extracts leaves (**a**), stems (**b**), and roots (**c**) of plants *Reynoutria japonica* and leaves (**d**), stems (**e**), and roots (**f**) of plants *Reynoutria sachalinensis*. 1—chlorogenic acid; 2*—coumaroyl quinic acid; 3—5-O-caffeoylquinic acid; 4—procyanidin dimer, type B; 5—epicatechin; 6*—coumaroyl quinic acid; 7—catechin; 8—piceatannol 3′-O-glucoside; 9—resveratroloside; 10*—coumaroyl quinic acid; 11—luteolin 7-O-glucoside; 12—trans-polydatin; 13—epicatechin-3-gallate; 14—quercetin-3-O-glucoside; 15—quercetin-3-O-pentoside; 16—quercitrin; 17—quercetin-3-O-pentoside; 18—cis-polydatin; 19—kaempferol-3-O-glucoside; 20—resveratrol; 21—quercetin; 22—torachrysone glucoside; 23—emodin glucoside; 24—emodin-8-O-(6′-O-malonyl)-d-glucoside; 25—hydropiperoside; 26—physcion-8-O-β-d-glucoside; 27—torachrysone; 28—vanicoside C; 29—vanicoside B; 30—questin; 31—emodin. *—Asterisk indicates isomer.

**Figure 3 plants-13-03330-f003:**
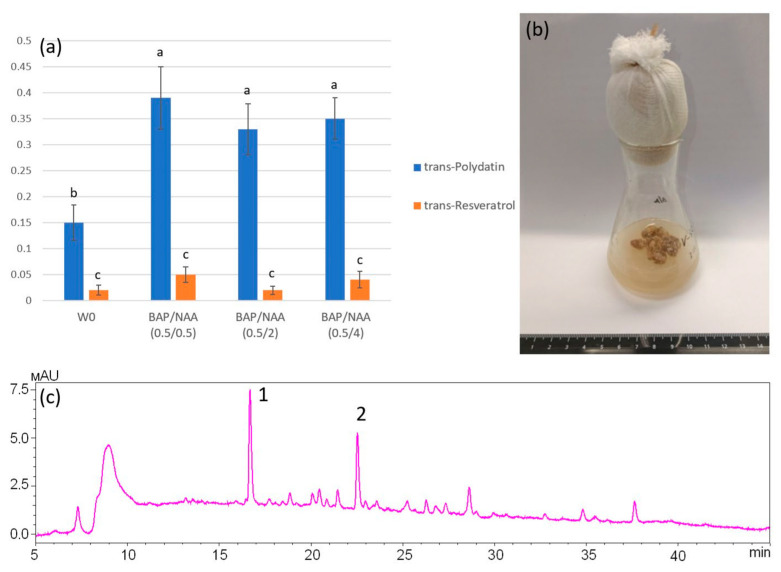
(**a**) Trans-polydatin and trans-resveratrol content in *Reynoutria japonica* cell culture. (**b**) Photograph of one-month-old *Reynoutria japonica* cell culture. (**c**) HPLC-UV chromatographic profiles (310 nm) of *Reynoutria japonica* cell culture. 1—*trans*-polydatin; 2—*trans*-resveratrol. Means on each column followed by the same letter were not different using ANOVA with Tukey’s pairwise comparisons.

**Table 1 plants-13-03330-t001:** Stilbenes content in leaves, stems, and roots of *Reynoutria japonica* and *Reynoutria sachalinensis*. Data are presented as mean ± standard error of the mean (SEM). Values in each row followed by the same letter were not different using one-way analysis of variance (ANOVA) (*p* < 0.05). tr.—trace amounts (less than 0.1 mg/g DW), n.d.—not detected.

Compounds (mg/g DW)	*R. japonica*			*R. sachalinensis*		
	Leaves	Stems	Roots	Leaves	Stems	Roots
Piceatannol 3′-O-glucoside	n.d.	n.d.	23.58 ± 0.915 ^a^	n.d.	n.d.	2.38 ± 0.169 ^b^
Resveratroloside	tr.	0.14 ± 0.026 ^c^	74.19 ± 2.878 ^a^	tr.	0.47 ± 0.058 ^d^	7.48 ± 0.533 ^b^
*trans*-polydatin	1.36 ± 0.102 ^e^	1.65 ± 0.259 ^e^	130.7187 ± 5.134 ^a^	6.48 ± 0.473 ^c^	3.41 ± 0.287 ^d^	17.06 ± 0.573 ^b^
*cis*-polydatin	n.d.	n.d.	n.d.	0.15 ± 0.029 ^a^	n.d.	n.d.
*trans*-resveratrol	tr.	0.16 ± 0.044 ^b^	0.68 ± 0.061 ^a^	0.15 ± 0.047 ^b^	0.07 ± 0.016 ^c^	0.12 ± 0.027 ^b^
Total content	1.36 ± 0.102 ^f^	1.94 ± 0.127 ^e^	229.17 ± 13.477 ^a^	6.79 ± 0.689 ^c^	3.94 ± 0.356 ^d^	27.04 ± 1.722 ^b^

**Table 2 plants-13-03330-t002:** Content of secondary metabolites in leaves, stems, and roots of *Reynoutria japonica* and *Reynoutria sachalinensis*. Data are presented as mean ± standard error of the mean (SEM). Values in each row followed by the same letter were not different using one-way analysis of variance (ANOVA) (*p* < 0.05). tr.—trace amounts (less than 0.1 mg/g DW), n.d.—not detected.

Compounds (mg/g DW)	*R. japonica*			*R. sachalinensis*		
	Leaves	Stems	Roots	Leaves	Stems	Roots
Chlorogenic acid	4.82 ± 1.03 ^a^	n.d.	n.d.	n.d.	n.d.	n.d.
Coumaroyl quinic acid	1.59 ± 0.376 ^a^	n.d.	n.d.	n.d.	n.d.	n.d.
5-O-Caffeoylquinic acid	2.03 ± 0.403 ^a^	n.d.	n.d.	n.d.	n.d.	n.d.
Procyanidin dimer, Type B	n.d.	n.d.	n.d.	n.d.	n.d.	0.48 ± 0.153 ^a^
Epicatechin	tr.	0.386 ± 0.106 ^a^	0.264 ± 0.03 ^a^	n.d.	n.d.	n.d.
Coumaroyl quinic acid	0.98 ± 0.249 ^a^	n.d.	n.d.	n.d.	n.d.	n.d.
Catechin	n.d.	n.d.	n.d.	3.62 ± 1.277 ^a^	n.d.	n.d.
Coumaroyl quinic acid	0.51 ± 0.096 ^a^	n.d.	n.d.	n.d.	n.d.	n.d.
Luteolin 7-O-glucoside	tr.	tr.	n.d.	0.2 ± 0.035 ^a^	tr.	n.d.
Epicatechin-3-gallate	n.d.	n.d.	n.d.	1.16 ± 0.311 ^a^	0.14 ± 0.048 ^b^	0.16 ± 0.040 ^b^
Quercetin-3-O-glucoside	n.d.	n.d.	n.d.	0.65 ± 0.03 ^a^	0.1 ± 0.022 ^b^	n.d.
Quercetin-3-O-pentoside	0.41 ± 0.027 ^b^	0.107 ± 0.026 ^c^	tr.	1.28 ± 0.329 ^a^	0.06 ± 0.009 ^c^	n.d.
Quercitrin	1.09 ± 0.193 ^b^	tr.	n.d.	13.96 ± 2.752 ^a^	0.41 ± 0.024 ^c^	n.d.
Quercetin-3-O-pentoside	6.59 ± 0.379 ^a^	0.191 ± 0.041 ^c^	tr.	tr.	0.83 ± 0.168 ^b^	n.d.
Kaempferol-3-O-glucoside	tr.	0.140 ± 0.096 ^b^	tr.	0.16 ± 0.026 ^b^	0.61 ± 0.142 ^a^	0.70 ± 0.025 ^a^
Quercetin	n.d.	n.d.	0.504 ± 0.031 ^a^	0.42 ± 0.096 ^a,b^	tr.	0.21 ± 0.051 ^b^
Torachrysone glucoside	n.d.	n.d.	n.d.	n.d.	n.d.	0.06 ± 0.025 ^a^
Emodin glucoside	2.47 ± 0.235 ^b^	tr.	9.71 ± 0.749 ^a^	1.18 ± 0.213 ^c^	0.54 ± 0.057 ^d^	0.81 ± 0.294 ^d^
Emodin-8-O-(6′-O-malonyl)-d- glucoside	n.d.	n.d.	5.22 ± 0.211 ^a^	n.d.	0.32 ± 0.042 ^c^	0.82 ± 0.149 ^b^
Hydropiperoside	n.d.	0.71 ± 0.275 ^a^	0.94 ± 0.063 ^a^	n.d.	0.66 ± 0.273 ^a^	0.76 ± 0.193 ^a^
Physcion-8-O-β-d-glucoside	n.d.	n.d.	n.d.	0.11 ± 0.029 ^a^	n.d.	n.d.
Torachrysone	n.d.	0.368 ± 0.132 ^b^	1.84 ± 0.28 ^a^	n.d.	0.52 ± 0.098 ^b^	2.09 ± 0.652 ^a^
Vanicoside C	n.d.	0.207 ± 0.082 ^c^	6.24 ± 0.597 ^a^	n.d.	0.59 ± 0.151 ^b^	6.49 ± 1.115 ^a^
Vanicoside B	n.d.	3.94 ± 1.02 ^c^	19.88 ± 1.259 ^a^	n.d.	14.34 ± 1.130 ^b^	21.53 ± 5.534 ^a^
Questin	n.d.	n.d.	n.d.	n.d.	tr.	1.36 ± 0.187 ^a^
Emodin	n.d.	n.d.	0.507 ± 0.131 ^a^	n.d.	n.d.	0.26 ± 0.112 ^b^

**Table 3 plants-13-03330-t003:** Fresh and dry biomass accumulation, total stilbene content, and stilbene production in cells cultures of *Reynoutria japonica* growing on different phytohormones. Data are presented as mean ± standard error (SE). Values in each column followed by the same letter were not different using one-way analysis of variance (ANOVA) (*p* < 0.05). W0—nutrient medium that does not contain phytohormones; BAP—6-benzylaminopurine; NAA—α-naphthaleneacetic acid.

Phytohormones	Fresh Weight (g/L)	Dry Weight (g/L)	Total Stilbene Content (mg/g DW)	Total Stilbene Production (mg/L)
W0	279.7 ± 38.8 ^b^	54.8 ± 6.6 ^c^	0.17 ± 0.07 ^b^	9.32 ± 1.13 ^c^
BAP/NAA (0.5/0.5 mg/L)	638.8 ± 51.8 ^a^	108.2 ± 8.2 ^a^	0.42 ± 0.11 ^a^	45.44 ± 9.85 ^a^
BAP/NAA (0.5/2 mg/L)	302.2 ± 38.7 ^b^	52.1 ± 6.8 ^c^	0.35 ± 0.05 ^a^	18.24 ± 3.55 ^b^
BAP/NAA (0.5/4 mg/L)	496.3 ± 57.1 ^a^	74.9 ± 6.2 ^b^	0.39 ± 0.10 ^a^	29.21 ± 1.98 ^a^

## Data Availability

Data are contained within the article and Appendix A.

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
