# Peer review of "Analysis of Phenolic Compounds of Reynoutria sachalinensis and Reynoutria japonica Growing in the Russian Far East"

_plants, 2024, doi:10.3390/plants13233330_

Round 1

Reviewer 1 Report

Comments and Suggestions for Authors

This manuscript systematically investigates the phylogenetics of Reynoutria japonica and Reynoutria sachalinensis in the Russian Far East, focusing on the content and composition of secondary metabolites in various tissue types, as well as callus culture. The study draws several conclusions that could support the development and utilization of Reynoutria in the Russian Far East. However, there are numerous issues within the manuscript that require clarification or revision.

1. It is unclear whether the manuscript examines the phenolic compounds and secondary metabolite composition of two samples or fourteen samples.

2. All species names in the manuscript need to be italicized.

3. The authors primarily provide molecular identification information for fourteen samples in the appendix, but they also need to conduct a homology comparison among these samples or construct a phylogenetic tree, along with relevant information (such as gene accession numbers and homology) for the species with the highest sequence similarity for each sample. Merely indicating in Figure 1 that samples 1-4 have the highest homology with R. sachalinensis and an evolutionary distance of 73 is insufficient.

4. Lines 118-125 require the provision of the described data.

5. The authors need to supply additional data, such as total phenolic content, total flavonoid content, and the antioxidant capacity of the extracts, or provide more data related to callus culture.

6. The data in Table 3 must include the results of significance testing.

7. A conclusion section needs to be added.

8. No details are provided in the methods section for section 2.3.

9. Line 376: Is it 9 plants or 10 plants?

10. The data in the abstract should be consistent with the results, such as stating 229.17 mg/g DW and 27.04 mg/g DW.

11. Photographs of the samples should be provided.

12. References for the methanol extraction method used in the manuscript should be included.

13. The title requires revision, as the current one does not adequately encapsulate the research content of the manuscript.

Author Response

  1. It is unclear whether the manuscript examines the phenolic compounds and secondary metabolite composition of two samples or fourteen samples.

Thank you very much for your attention. In this work, we investigated metabolites of plants of two species. R. sachalinensis plants were collected from 4 points distant from each other. In each point, 2 plants were collected. It is difficult to say whether these were separate plants, since Reinutria can grow strongly vegetatively. For this reason, plants from one point were considered a technical repeat. We also did the same for Reinutria plants collected in Primorsky krai. In total, we analyzed 8 plants collected from 4 points on Sakhalin and 10 plants collected from 5 points in Primorsky Krai. For each plant, the leaves, stem and root were analyzed for secondary metabolites. HPLC-MS analysis showed that all samples from Sakhalin Island had the same spectrum of metabolites, the amount varied slightly. Similar results were obtained for plants collected in Primorsky Krai. Therefore, the results from all points on Sakhalin and Primorsky Krai were combined and presented in the final Tables 1 and 2. Here we tried to make things clearer, and we also added an explanation of Lines 144-147.

  1. All species names in the manuscript need to be italicized.

Thank you very much for noticing this. Apparently, some of the italics disappeared during editing of the manuscript. We carefully reread the entire text and corrected this defect.

  1. The authors primarily provide molecular identification information for fourteen samples in the appendix, but they also need to conduct a homology comparison among these samples or construct a phylogenetic tree, along with relevant information (such as gene accession numbers and homology) for the species with the highest sequence similarity for each sample. Merely indicating in Figure 1 that samples 1-4 have the highest homology with R. sachalinensis and an evolutionary distance of 73 is insufficient.

Thanks to the reviewer for the interesting question. We have tried to answer it here and have also added information to Lines 104-119 and Figure S2.

We have re-examined these results and performed additional analysis and description. Specifically, the matK and rps16 gene sequence analysis revealed that samples 1-4 were completely identical to R. sachalinensis, while samples 5-14 corresponded to R. japonica (see Figure S2). Further analysis of the ITS sequences revealed that samples 5-14 were also completely identical to R. japonica. However, the ITS sequences of samples 1-4 differed significantly from the previously described ITS sequences for R. sachalinensis.

This phenomenon can be explained by intraspecific variability of the ITS gene, indicating possible genetic differences within this species. Note that the available R. sachalinensis ITS sequences in the Genbank were collected from plants growing at a significant distance from the samples we studied, for example, from Korea or Europe. This may indicate geographic variability within the species. It is also possible that we encountered a subspecies or variation of R. sachalinensis in Sakhalin Oblast, which requires further detailed study.

  1. Lines 118-125 require the provision of the described data.

Thank you very much. We have tried to bring more clarity to this issue.

  1. The authors need to supply additional data, such as total phenolic content, total flavonoid content, and the antioxidant capacity of the extracts, or provide more data related to callus culture.

Thank you very much for the recommendation, it is very interesting. The aim of this work was to identify the qualitative and quantitative composition of secondary metabolites using HPLC-MS/MS data. Mainly major peaks of substances, the identification of which could be performed reliably, were taken into account. The obtained results allow the reader to estimate the contribution of individual or groups of substances to the total pool of identified compounds. Analysis of total phenolic compounds using the Folin-Ciocolteu or FBBB methods was considered inappropriate, since the methods do not have absolute selectivity (doi:10.1016/J.LWT.2024.115756). The inevitable content of some substances, such as sugars, ascorbic acid, some protein compounds and amino acids may contribute, distorting the true results of the phenolic compounds content, thus the obtained results can only be assessed as preliminary. Evaluation of total antioxidant activity was not the aim of this work.

  1. The data in Table 3 must include the results of significance testing.

Thank you very much for the remark, indeed, it was worth doing earlier. We added an assessment of the significance of values ​​using one-way analysis of variance (ANOVA) (p < 0.05). You can evaluate the work done in Table 3.

  1. A conclusion section needs to be added.

Thank you, we have added this section.

In conclusion, it is shown that there is a significant difference in the composition of metabolites between R. japonica and R. sachalinensis plants. The root of R. japonica is a richer source of stilbenes, while the leaves of R. sachalinensis contain a higher concentration of quercetin derivatives. These and other differences in the metabolic profile may indicate the adaptation of each species to its habitat. Moreover, they emphasize the possibility of using the detected metabolites as biomarkers for species identification. In turn, the presence of common metabolites may indicate a genetic proximity between R. japonica and R. sachalinensis. However, despite fast growth and dangerous as a weed, these Reynoutria plants are used used in many areas of human activity, such as medicine, cosmetology, agriculture and even cooking, which means that research aimed at studying these plants requires continuation.

  1. No details are provided in the methods section for section 2.3.

Thank you very much for the advice, we have taken it into account. For section 2.3. Stilbene accumulation in cell cultures of R. japonica, we have added information on the methodological part in lines 423-426.

  1. Line 376: Is it 9 plants or 10 plants?

We have clarified the results, namely 10 plants of R. japonica were collected. The updated data is presented in Line 401.

  1. The data in the abstract should be consistent with the results, such as stating 229.17 mg/g DW and 27.04 mg/g DW.

Thank you for your recommendation, we have taken it into account. Previously, we wanted to focus the reader's attention on the whole number, since it is easier to perceive in the abstract, but we understand your point of view and support it. The abstract contains corrected data.

  1. Photographs of the samples should be provided.

Thank you for the recommendation. When preparing the manuscript, we wanted to include a photo of the plants, but unexpected difficulties arose. The fact is that different people participated in collecting the material, and they did not receive recommendations on photographing the samples. As a result, we received different types of photographs, which do not look presentable enough in a single drawing. Undoubtedly, this is a stupid mistake in planning the experiment, but we took it into account. However, we presented the Figure containing the photo in the supplementary materials (Figure S3).

  1. References for the methanol extraction method used in the manuscript should be included.

Thanks for the recommendation. We have added the necessary information (Line 452).

  1. The title requires revision, as the current one does not adequately encapsulate the research content of the manuscript.

Thank you for your recommendation. We appreciate your concern, but we have consulted with the writing team and are also listening to the second reviewer that the manuscript title is appropriate. Of course, there is no perfect title. One can try to cover all the results of the study, but we chose to focus on the main purpose of the study and reflect it in the title.

Reviewer 2 Report

Comments and Suggestions for Authors

A manuscript entitled "Analysis of Phenolic Compounds of Reynoutria sachalinensis 2 and Reynoutria japonica Growing in the Russian Far East" has been submitted to Plants journal in Plant Physiology and Metabolism section. 

The paper is well written, the experiment is very clearly presented, together with the supplementary files, so that all information can be found from the very beginning, which shows a lot of effort in design and verification of the methodology. The references also cover the topic from the inside out, it shows that the authors know the topic and are specialists in it. The title is informative, the same for abstract.

The first remark would be to add more various applications of these two plant extracts, not only pharmaceutical but also there is a biopesticide utylization possibility although Reynoutria extract has not been registered in Europe yet but demonstrates good alternative and is registered in US. Could you elucidate using your results? Do you see any threats for agricultural sector? 

My second remark would be about the methodology. Why other authors from references revealed more compounds? More than 150? Is this because Reynoutria differs so much depending on the region it grows or you had some limitations connected with the eqipment used? Or is there any other explanation?

There is HPLC-MC in the keywords section, shouldn't it be HPLC-MS? 

Author Response

  1. The first remark would be to add more various applications of these two plant extracts, not only pharmaceutical but also there is a biopesticide utylization possibility although Reynoutria extract has not been registered in Europe yet but demonstrates good alternative and is registered in US. Could you elucidate using your results? Do you see any threats for agricultural sector? 

Thank you very much for your advice, this is really very interesting. We have studied the literature and added information about the use of Reynoutria as a biopesticide, as well as a plant growth and development stimulant. This information is presented in Lines 76-85. Of course, Reynoutria extracts can be used in agriculture and this is an interesting idea for future research.

  1. My second remark would be about the methodology. Why other authors from references revealed more compounds? More than 150? Is this because Reynoutria differs so much depending on the region it grows or you had some limitations connected with the eqipment used? Or is there any other explanation?

In this study, we focused on the main substances, the amount of which was greater than 0.1 mg/g DW. On the one hand, it allows us to focus on the study of the main metabolites, but on the other hand, we also cannot confidently identify minor substances due to insufficient resolution of the mass spectrometer. Moreover, we tried to give not only a qualitative but also a quantitative assessment of metabolites. Minor compounds are difficult to quantify, since they can often look like noise on the chromatogram. Also, for quantitative determination, we would need additional analytical standards of substances. Of course, it is undeniable that metabolism is affected by a huge number of factors, including the place and conditions of growth. However, according to literary data, the main substances and their quantities are still comparable with plants growing in different areas.

  1. There is HPLC-MC in the keywords section, shouldn't it be HPLC-MS? 

We apologize for this typo. Thank you for pointing it out to us.

Round 2

Reviewer 1 Report

Comments and Suggestions for Authors

The author has responded to my comments.

1. The author responded to comment 1 with, “In total, we analyzed 8 plants collected from 4 locations on Sakhalin and 10 plants collected from 5 locations in Primorsky Krai.” Consequently, lines 403-404 have been amended to reflect 10 and 8. Which specific 14 samples are indicated in various sections of Figure 1 and the manuscript?

2. Is the homology between the two categories of samples consistently 100%? If not, please provide the specific data.

3. The title emphasizes the key research focus rather than providing a summary; however, the abstract only describes the results from section 2.2, with no mention of the content from sections 2.1 and 2.3. Similarly, the discussion does not address the content of section 2.1.

4. The abstract states, “This result may indicate the adaptation of plants to their habitat and can also be used as biomarkers for species identification. In turn, the presence of common metabolites may indicate genetic proximity between species.” Please provide the corresponding data or detailed explanations that support these conclusions. This content may be more appropriate for the discussion rather than the abstract.

5. Table 3 indicates that the growth of cell cultures under hormone treatment is significantly enhanced; however, while the stilbene content has significantly increased compared to the control, the stilbene levels remain quite low. More discussion is needed here, including current research advancements and the effects of these two hormones on other cultures. Additionally, the rationale for the concentrations of hormones selected in the manuscript should be provided, and whether the experimental design requires further refinement.

6. In my opinion, the research content in this manuscript is relatively limited and insufficient to support a research paper. I recommend that the author provide additional relevant research data, such as images related to cell culture and more comprehensive data.

Author Response

Dear Reviewer, thank you for your sincere desire to help us present our manuscript better, we really appreciate it. Below we have provided answers to your questions and comments.

  1. The author responded to comment 1 with, “In total, we analyzed 8 plants collected from 4 locations on Sakhalin and 10 plants collected from 5 locations in Primorsky Krai.” Consequently, lines 403-404 have been amended to reflect 10 and 8. Which specific 14 samples are indicated in various sections of Figure 1 and the manuscript?

For DNA extraction, we used leaves of 4 R. sachalinensis plants (Sample:1-4) and 10 R. japonica plants (Sample:5-14).

For HPLC analysis, we used 8 R. sachalinensis and 10 R. japonica plants, the HPLC results were summarized in Table 1,2. (Lines 466-467).

  1. Is the homology between the two categories of samples consistently 100%? If not, please provide the specific data.

The sequences of the studied genes of samples 1-4 were completely identical to each other, and samples 5-14 were identical to each other. However, not all sequences were 100% identical to the previously described genes. We also made an additional Figure S2, which shows in detail all identical and different regions of the sequences of the studied genes. Moreover, we compared all the sequences and presented the percentage of their homology to each other, as well as to the previously described sequences of the rps16, matK and ITS2 genes of R. japonica and R. sachalinensis. We have also improved the Phylogenetic analysis section (Lines 105-123). Also, all nucleotide sequences are presented in Table S5.

  1. The title emphasizes the key research focus rather than providing a summary; however, the abstract only describes the results from section 2.2, with no mention of the content from sections 2.1 and 2.3. Similarly, the discussion does not address the content of section 2.1.

Thanks again. If you suggest a more appropriate title, we may consider it. Also following your recommendation, we have added a short summary of the results from sections 2.1 and 2.3 to the abstract (lines 13-15 and 30-31). A discussion of the results from section 2.1 has been added to the discussion (lines 281-301).

  1. The abstract states, “This result may indicate the adaptation of plants to their habitat and can also be used as biomarkers for species identification. In turn, the presence of common metabolites may indicate genetic proximity between species.” Please provide the corresponding data or detailed explanations that support these conclusions. This content may be more appropriate for the discussion rather than the abstract.

Thank you for your comment. Indeed, this statement may not be entirely appropriate for this section. Based on this, we have removed it from this section.

  1. Table 3 indicates that the growth of cell cultures under hormone treatment is significantly enhanced; however, while the stilbene content has significantly increased compared to the control, the stilbene levels remain quite low. More discussion is needed here, including current research advancements and the effects of these two hormones on other cultures. Additionally, the rationale for the concentrations of hormones selected in the manuscript should be provided, and whether the experimental design requires further refinement.

Thank you very much for the suggestion. At this point we understand that it would have been possible to study the effects of various hormones on cell culture in more detail, but this requires a separate study. These hormones were chosen based on information in the literature, in particular:

  • Wen T, Liang L, Zeng Y, Yu X, “Effect of different light intensity on Polygonum cuspidatum callus”, China Journal of Chinese Materia Medica, vol. 32(13), pp. 1277-1280, 2007.
  • Ou J Q, Chen X X, Ding L H, Cao Y, “Callus Inducement and Resveratrol Formation of Polygonum cuspidatum”, Journal of Central South Forestry University, 26(3), pp. 24-27, 2006.
  • Li, X.L.; Wang, D.D.; Gao, J.; Cui, X.H.; Zhao, J.; Zhang, L.M.; Gao, W.Y.; Xiao, L. Preliminary Study on Inducing Incompact Callus of Polygonum Cuspidatum Using the Medium with Perlite. AMR 2011, 365, 140–144, doi:10.4028/www.scientific.net/AMR.365.140.
  • And also a patent: CN104372034A «Method for production of resveratrol from polygonum cuspidatum trichoid root and enlarged cultivation».

We have also added this information to the discussion section (Lines 430-434).

  1. In my opinion, the research content in this manuscript is relatively limited and insufficient to support a research paper. I recommend that the author provide additional relevant research data, such as images related to cell culture and more comprehensive data.

Thank you very much. We agree with you. And following your recommendations, we added data such as individual stilbenes content in cell culture, a photograph of the cell culture, and HPLC-UV chromatographic profiles of Reynoutria japonica cell culture (see Figure 3). Lines 251-253.

Round 3

Reviewer 1 Report

Comments and Suggestions for Authors

The author has addressed all of my comments and made revisions to the manuscript.